# Fecal Calprotectin as a Prognostic Biomarker for Mortality and Renal Outcomes in Chronic Kidney Disease

**DOI:** 10.3390/biom15040557

**Published:** 2025-04-10

**Authors:** So Young Lee, Kyungdo Han, Hyuk-Sang Kwon, Eun Sil Koh, Sungjin Chung

**Affiliations:** 1Division of Nephrology, Department of Internal Medicine, Yeouido St. Mary’s Hospital, College of Medicine, The Catholic University of Korea, Seoul 07345, Republic of Korea; leesoyoung@catholic.ac.kr; 2Department of Statistics and Actuarial Science, Soongsil University, Seoul 06978, Republic of Korea; hkd@ssu.ac.kr; 3Division of Endocrinology and Metabolism, Department of Internal Medicine, Yeouido St. Mary’s Hospital, College of Medicine, The Catholic University of Korea, Seoul 07345, Republic of Korea; drkwon@catholic.ac.kr

**Keywords:** fecal calprotectin, biomarker, chronic kidney disease, mortality, renal function, systemic inflammation, cardiovascular disease, gut–kidney axis

## Abstract

Background/Objectives: Fecal calprotectin (FC) is a biomarker of intestinal inflammation widely used in the assessment of gastrointestinal disorders. However, its role in chronic kidney disease (CKD) remains unclear. Given the growing recognition of the gut–kidney axis in CKD pathophysiology, this study aimed to investigate the association between FC levels, systemic inflammation, renal outcomes, and mortality in CKD patients. Methods: We enrolled a total of 515 CKD patients who underwent fecal calprotectin measurement between 2016 and 2023. After applying the exclusion criteria (inflammatory bowel disease, ongoing renal replacement therapy, or incomplete laboratory data), 260 patients were included in the final analysis and stratified into low-FC (<102 μg/g, *n* = 130) and high-FC (≥102 μg/g, *n* = 130) groups based on the median FC value. Factors associated with kidney disease progression and patient survival were analyzed. Results: Patients in the high-FC group (≥102 μg/g) were significantly older (72.8 ± 14.63 vs. 64.02 ± 18.15 years, *p* < 0.0001) and had a higher prevalence of diabetes mellitus (55.38% vs. 42.31%, *p* = 0.0349), heart failure (21.54% vs. 7.69%, *p* = 0.0016), and history of acute kidney injury (33.85% vs. 18.46%, *p* = 0.0048). Elevated FC was independently associated with increased mortality risk (hazards ratio [HR] 1.658, 95% confidence interval [CI] 1.034–2.658, *p* = 0.0357) with higher mortality rates (48.36 vs. 18.46 per 100,000 person-years). Subgroup analyses revealed stronger associations between FC and mortality in males (HR 2.160, 95% CI 1.046–4.463, *p* = 0.0375), elderly patients (≥75 years) (HR 2.122, 95% CI 1.209–3.725, *p* = 0.0088), and non-diabetic patients (HR 2.487, 95% CI 1.141–5.421, *p* = 0.0219). While FC was not significantly associated with end-stage kidney disease (ESKD) progression (odds ratio [OR] 1.289, 95% CI 0.455–3.650, *p* = 0.6323), higher FC levels paradoxically predicted slower estimated glomerular filtration rate (eGFR) decline (OR 2.763, 95% CI 1.139–6.699, *p* = 0.0245). Combined analysis revealed patients with both elevated FC and high-sensitivity C-reactive protein (hs-CRP) had the highest mortality risk (HR 3.504, 95% CI 1.163–10.554, *p* < 0.0001) compared to those with low levels of both markers. Conclusions: FC is a potential prognostic biomarker for mortality in CKD patients, independently of traditional inflammatory markers. Further research is warranted to elucidate the mechanisms underlying its paradoxical relationship with renal outcomes and its potential role in risk stratification and therapeutic targeting in CKD.

## 1. Introduction

Chronic kidney disease (CKD) is a major global health concern, affecting approximately 10–13.4% of the world’s population—over 800 million individuals [1,2]. Its prevalence increased by 33% between 1990 and 2017, and in 2019, more than 3.1 million deaths were attributed to impaired kidney function [3]. CKD is projected to become the fifth leading cause of years of life lost (YLLs) globally by 2040 [4]. It is commonly associated with cardiovascular disease, anemia, and metabolic complications, which contribute to diminished quality of life and increased healthcare utilization [3]. Despite its clinical burden, early-stage CKD is often asymptomatic and underdiagnosed, underscoring the need for reliable and non-invasive biomarkers to enable early risk stratification and prognosis.

Calprotectin, a 36.5 kDa calcium- and zinc-binding protein complex comprising the S100A8/S100A9 heterodimer, constitutes approximately 60% of neutrophil cytoplasmic proteins and is primarily found in plasma, urine, cerebrospinal fluid, stool, and saliva. This multifunctional protein is predominantly expressed in neutrophils and monocytes, where it acts as a damage-associated molecular pattern (DAMP) molecule, playing crucial roles in innate immunity, inflammatory responses, and cellular homeostasis. Through the activation of pattern recognition receptors, particularly Toll-like receptor 4 (TLR4), calprotectin amplifies inflammatory cascades and modulates immune responses [5,6].

The clinical utility of calprotectin extends across various kidney diseases, with distinct patterns of expression in different body fluids. In acute kidney injury (AKI), urinary calprotectin has emerged as a promising early diagnostic biomarker, demonstrating superior performance in distinguishing between pre-renal and intrinsic AKI compared to traditional markers [7,8,9]. Notably, urinary calprotectin levels increase more rapidly than established biomarkers such as neutrophil gelatinase-associated lipocalin (NGAL) and kidney injury molecule-1 (KIM-1), potentially enabling earlier therapeutic interventions [10,11,12,13]. In CKD, both serum and fecal calprotectin (FC) serve as clinically relevant markers. Recent longitudinal studies have demonstrated that serum calprotectin independently predicts cardiovascular outcomes in CKD patients, with elevated levels associated with increased mortality risk over 5–10 years of follow-up [14,15,16]. Furthermore, calprotectin has been identified as a potential therapeutic target for vascular calcification in CKD, highlighting its pathogenic role beyond biomarker status [16,17].

While serum, plasma, and urinary calprotectin have been extensively studied in kidney diseases, recent studies have also highlighted gut-derived inflammation as a potential driver of systemic inflammation in CKD. FC is a well-established non-invasive biomarker of intestinal inflammation, particularly in conditions such as inflammatory bowel disease, with levels directly proportional to neutrophil migration into the gastrointestinal tract [18]. With the increasing recognition of the gut–kidney axis, FC has emerged as a novel marker that may provide additional insights into intestinal immune activation, disease progression and patient outcomes. Its stability at room temperature, resistance to bacterial degradation, and uniform distribution in feces further support its clinical utility [19].

Recent research has established a bidirectional interaction between dysbiotic gut microbiota and impaired renal function in CKD pathogenesis. As renal function declines, uremia alters the gut’s microbial composition and reduces the epithelial barrier’s integrity, enabling microbial translocation and systemic inflammation. Conversely, microbial metabolites and uremic toxins originating in the gut contribute to systemic inflammation, kidney injury, and cardiovascular complications [20,21]. Gut dysbiosis in CKD is characterized by reduced microbial diversity and a shift toward urease- and indole-producing bacteria [22,23,24,25,26], leading to increased production of protein-bound uremic toxins such as indoxyl sulfate and p-cresyl sulfate [27,28,29]. These toxins promote oxidative stress, endothelial dysfunction, and renal fibrosis, accelerating CKD progression [22,23,24,25,26,30,31,32,33]. Additionally, microbial metabolites—including trimethylamine N-oxide (TMAO), phenylacetylglutamine, and hippuric acid—have been implicated in both renal and cardiovascular complications [27,34]. Gut-derived metabolites, particularly TMAO and indole derivatives, have been linked to vascular calcification, endothelial dysfunction, and increased cardiovascular mortality, the leading cause of death in CKD [27,35,36]. These findings support the concept of a “gut–kidney–cardiovascular axis”, wherein gut dysbiosis and its metabolic byproducts mediate systemic complications beyond the renal system. This concept highlights the importance of identifying reliable biomarkers that reflect gut-derived toxicity and systemic inflammatory burden in CKD.

Despite accumulating evidence that CKD is associated with intestinal inflammation and dysbiosis, the prognostic significance of FC in CKD patients remains unclear. While several studies have demonstrated elevated FC levels in CKD [37,38,39], its relationship with disease progression, inflammation, and clinical outcomes has not been systematically evaluated. Traditional inflammatory markers such as C-reactive protein (CRP) are widely used in CKD [14,16], yet they may not adequately reflect local intestinal inflammation [40]. FC, as a more specific marker of gut-derived inflammation, may offer additional insights into CKD pathophysiology.

This study aimed to investigate whether fecal calprotectin serves as a reliable biomarker of systemic inflammation in CKD patients. Specifically, we will evaluate its correlation with established inflammatory markers, such as CRP, and assess its prognostic value in predicting CKD progression and mortality.

## 2. Materials and Methods

### 2.1. Study Design and Population

We conducted a cross-sectional study of CKD patients admitted to the Department of Nephrology at Yeouido St. Mary’s Hospital between March 2016 and April 2023. The study was approved by the hospital’s Ethics Committee (approval number: SC23RISI0098).

A total of 515 hospitalized patients, whose fecal calprotectin levels were measured during their admission, were initially screened for inclusion. Patients were excluded if they met any of the following criteria: (1) diagnosis of inflammatory bowel disease (IBD) at the time of enrollment, (2) ongoing renal replacement therapy (RRT) for end-stage kidney disease (ESKD), or (3) incomplete laboratory data, specifically missing urine or blood test results. After these exclusion criteria were applied, 260 patients were included in the final analysis. Participants were categorized into two groups based on the median FC value: the low-FC group (<102 μg/g, *n* = 130) and the high-FC group (≥102 μg/g, *n* = 130).

### 2.2. Data Collection

We collected comprehensive demographic and clinical data from the hospital information system (HIS), including age, sex, and comorbidities (diabetes, hypertension, malignancy, glomerular disease, heart failure, coronary artery disease, chronic liver disease, and history of AKI). Medication history was also obtained, with special attention given to drugs that might influence gut inflammation or microbiota composition (e.g., polystyrene sulfonate, ferrous sulfate, AST-120).

Laboratory parameters were obtained from standardized blood, fecal, and urine samples. Blood parameters included hemoglobin (Hb), hematocrit (Hct), blood urea nitrogen (BUN), serum creatinine (Cr), estimated glomerular filtration rate (eGFR, mL/min/1.73 m^2^, calculated using the CKD-EPI equation), albumin, and electrolytes. Urine parameters included 24 h urine volume and protein excretion.

FC levels were quantified using an immunoenzymatic assay and measured with an ${ichroma}^{TM}$ III Calprotectin analyzer (Boditech Med, Inc., Chuncheon, Republic of Korea). The reference range for FC was <50 μg/g feces, with a detection range of 10–1000 μg/g. All samples were processed within 24 h of collection.

Regarding study outcomes, we analyzed all-cause mortality and the initiation of RRT, including kidney transplantation (KT), hemodialysis (HD), and peritoneal dialysis (PD). Additionally, to assess renal function decline at follow-up, we recorded the most recent serum creatinine and eGFR values and analyzed the rate of change in eGFR, calculated as follows: (last eGFR—initial eGFR)/initial eGFR.

### 2.3. Statistical Analysis

Descriptive statistics were presented as mean ± standard deviation (SD) for normally distributed continuous variables and as median (interquartile range, IQR) for non-normally distributed variables. Categorical variables were expressed as frequencies and percentages.

Predictors of renal outcomes were evaluated using logistic regression analysis. To assess factors associated with all-cause mortality, we performed Cox proportional hazards regression analysis, focusing on whether FC, independently or in combination with serum high-sensitivity C-reactive protein (hs-CRP), could predict patient survival. Kaplan–Meier survival curves were constructed.

All the analyses were conducted using IBM SPSS Statistics version 25.0 (IBM Corp., Armonk, NY, USA) and R software version 4.1.0 (R Foundation for Statistical Computing, Vienna, Austria). A *p*-value < 0.05 was considered statistically significant.

## 3. Results

### 3.1. Basal Characteristics of the Study Population

Of the 515 CKD patients initially screened, 260 met the inclusion criteria and were included in the final analysis. The mean FC level was 136.22 ± 112.84 μg/g, with a median of 102 μg/g (IQR 30–300 μg/g). Patients were stratified into two groups based on the median FC value: low-FC group (<102 μg/g, *n* = 130) and high-FC group (≥102 μg/g, *n* = 130).

As summarized in Table 1, patients in the high-FC group were significantly older (72.8 ± 14.63 vs. 64.02 ± 18.15 years, *p* < 0.0001) and had a higher prevalence of diabetes mellitus (55.38% vs. 42.31%, *p* = 0.0349), heart failure (21.54% vs. 7.69%, *p* = 0.0016), and history of acute kidney injury (33.85% vs. 18.46%, *p* = 0.0048). Conversely, glomerular diseases were more prevalent in the low-FC group (23.08% vs. 9.23%, *p* = 0.0024).

Laboratory findings revealed significantly higher hs-CRP levels in the high-FC group [median 18.6 mg/L (IQR 3.33–86.61) vs. 2.61 mg/L (IQR 0.67–13.24), *p* = 0.0032]. Patients in the high-FC group also had lower hemoglobin (10.39 ± 2.2 vs. 11.15 ± 2.32 g/dL, *p* = 0.0073), hematocrit (30.3 ± 6.32% vs. 32.61 ± 6.6%, *p* = 0.0043), total protein (6.0 ± 0.91 vs. 6.41 ± 0.92 g/dL, *p* = 0.0003), albumin (3.19 ± 0.7 vs. 3.55 ± 0.69 g/dL, *p* < 0.0001), and potassium levels (4.09 ± 0.79 vs. 4.31 ± 0.89 mmol/L, *p* = 0.032).

Notably, occult blood in stool was significantly more common in the high-FC group (19.23% vs. 1.54%, *p* < 0.0001). However, no significant differences were observed between groups regarding sex distribution, prevalence of hypertension, malignancy, coronary artery disease, chronic liver disease, medication use, renal function parameters (BUN, serum creatinine, eGFR), sodium levels, and urinary protein excretion.

### 3.2. Renal Outcomes

During the study period, 119 patients (45.8%) progressed to ESKD, requiring renal replacement therapy. Multivariate logistic regression analysis identified several independent predictors of ESKD development. Higher hemoglobin (odds ratio (OR) 6.034, 95% confidence interval (CI) 1.719–21.183, *p* = 0.005), higher blood urea nitrogen (OR 1.034, 95% CI 1.009–1.06, *p* = 0.0076), higher total protein (OR 2.393, 95% CI 1.312–4.366, *p* = 0.0045), higher hs-CRP (OR 1.475, 95% CI 1.123–1.936, *p* = 0.0052), and higher urine volume levels (OR 1.001, 95% CI 1.000–1.001, *p* = 0.0248) were associated with an increased risk of ESKD development. Conversely, higher hematocrit (OR 0.611, 95% CI 0.401–0.930, *p* = 0.0217) and higher serum creatinine levels (OR 0.211, 95% CI 0.121–0.368, *p* < 0.0001) were associated with a lower risk of ESKD progression. Importantly, fecal calprotectin levels were not significantly associated with ESKD development (OR 1.289, 95% CI 0.455–3.650, *p* = 0.6323). The details are presented in Table 2.

To further explore renal function changes, we analyzed predictors of the eGFR change rate in 174 patients who had both baseline and follow-up eGFR measurements (ranging from 0 to 150 mL/min/1.73 m^2^). The eGFR change rate was defined as the highest quartile (Q3) of eGFR change, calculated as (last_eGFR–initial_eGFR)/last_eGFR. This definition indicates that a higher eGFR change rate is associated with a lower risk of renal function deterioration.

As shown in Table 3, elevated FC levels were associated with a reduced risk of renal function decline (OR 2.763, 95% CI 1.139–6.699, *p* = 0.0245) among patients with serial eGFR measurements. Conversely, patients with a history of AKI (OR 0.295, 95% CI 0.131–0.666, *p* = 0.0033) and higher hs-CRP levels (OR 0.605, 95% CI 0.483–0.757, *p* < 0.0001) exhibited an increased risk of renal function deterioration.

### 3.3. Survival Outcomes

During the follow-up period, patients with higher FC levels had significantly higher mortality rates (48.36 vs. 18.46 per 100,000 person-years).

Cox proportional hazards regression analysis revealed that elevated FC levels were independently associated with increased mortality risk (hazards ratio (HR) 1.658, 95% CI 1.034–2.658, *p* = 0.0357) after adjusting for other clinical variables. Additional significant predictors of increased mortality included advanced age (HR 1.061, 95% CI 1.034–1.088, *p* < 0.0001), the presence of malignancy (HR 1.872, 95% CI 1.113–3.147, *p* = 0.0179), coronary artery disease (HR 3.506, 95% CI 1.739–7.071, *p* = 0.0004), higher blood urea nitrogen (HR 1.018, 95% CI 1.007–1.029, *p* = 0.0012), and higher total protein (HR 1.592, 95% CI 1.148–2.208, *p* = 0.0052).

Conversely, higher serum albumin levels (HR 0.304, 95% CI 0.182–0.507, *p* < 0.0001), higher serum creatinine levels (HR 0.800, 95% CI 0.669–0.956, *p* = 0.0141), and a history of AKI (HR 0.550, 95% CI 0.327–0.924, *p* = 0.0241) were associated with a reduced risk of mortality. Hs-CRP showed a trend toward increased mortality risk, but it did not reach statistical significance (HR 1.131, 95% CI 0.979–1.305, *p* = 0.0926) (Figure 1 and Table 4).

We conducted subgroup analyses to assess whether the association between elevated FC levels and mortality varied by sex, age, and diabetes status. All analyses were adjusted for age, albumin, malignancy, total protein, coronary artery disease, blood urea nitrogen, creatinine, history of AKI, and hs-CRP (Figure 2 and Table 5).

#### 3.3.1. Sex-Specific Analysis

In male patients, elevated FC levels were significantly associated with increased mortality risk (HR 2.160, 95% CI 1.046–4.463, *p* = 0.0375), with incidence rates of 47.62 vs. 17.87 per 100,000 person-years in the high- vs. low-FC groups. In contrast, no significant association was observed in female patients (HR 1.301, 95% CI 0.667–2.538, *p* = 0.4403).

#### 3.3.2. Age-Specific Analysis

The association between FC and mortality was age-dependent. In elderly patients (≥75 years), elevated FC levels were significantly associated with increased mortality (HR 2.122, 95% CI 1.209–3.725, *p* = 0.0088), with mortality rates of 92.83 vs. 49.16 per 100,000 person-years in the high- vs. low-FC groups. However, no significant association was observed in patients below 75 years (HR 0.828, 95% CI 0.224–3.066, *p* = 0.7772), for which, interestingly, the incidence rate was lower in the high- FC group (5.62 vs. 14.45 per 100,000 person-years).

#### 3.3.3. Diabetes Status-Specific Analysis

In non-diabetic patients, elevated FC levels were significantly associated with increased mortality risk (HR 2.487, 95% CI 1.141–5.421, *p* = 0.0219), with incidence rates of 46.80 vs. 12.3 per 100,000 person-years. However, in diabetic patients, this association was not significant (HR 1.194, 95% CI 0.603–2.362, *p* = 0.6110), despite higher overall mortality rates in both FC groups (49.66 vs. 27.33 per 100,000 person-years).

#### 3.3.4. Combined Effect of FC and CRP on Mortality

To further examine the prognostic significance of FC, we analyzed its combined effect with CRP on mortality risk. FC levels were categorized into three groups: low (FC < 30 µg/g), medium (30 µg/g ≤ FC < 300 µg/g), and high (FC ≥ 300 µg/g). C-reactive protein levels were divided into two groups: low (CRP < 5 mg/L) and high (CRP ≥ 5 mg/L). Patients were stratified into six inflammatory profile groups based on FC and CRP levels.

The combined FC and CRP inflammatory profiles were significantly associated with mortality (*p* < 0.0001). Patients with both high FC and high CRP levels had the highest mortality risk (HR 3.504, 95% CI 1.163–10.554), with an incidence rate of 69.9621 per 100,000 person-years. Notably, patients with low FC but high CRP levels showed a trend toward reduced mortality risk (HR 0.383, 95% CI 0.079–1.86) compared to the reference group. Additionally, the low-FC and high-CRP group demonstrated significantly lower mortality incidence rates (11.49 per 100,000 person- years) compared to those with medium FC and low CRP levels (18.53) or high FC and low CRP levels (40.22), suggesting that FC may provide additional prognostic information beyond CRP alone (Figure 3 and Table 6).

## 4. Discussion

In this study, we investigated the potential role of FC as a biomarker of systemic inflammation in CKD and its association with renal outcomes and mortality. Our analysis of 260 patients yielded several key findings. Patients with elevated FC levels (≥102 μg/g) exhibited distinct clinical profiles, including older age, a higher prevalence of diabetes mellitus and heart failure, and increased inflammatory markers, such as hs-CRP. Although FC was not predictive of ESKD progression, it was unexpectedly associated with a slower rate of eGFR decline. Notably, elevated FC levels were independently associated with increased all-cause mortality, particularly in males, elderly patients (≥75 years), and non-diabetic CKD patients. Furthermore, our combined assessment of FC and CRP revealed that patients with both elevated FC and CRP levels had the highest mortality risk, whereas those with low FC but high CRP levels exhibited a trend toward reduced mortality risk. These findings highlight the complex and multifactorial relationship between intestinal inflammation, systemic inflammatory burden, and mortality, reinforcing the growing relevance of the gut–kidney axis (Figure 4).

One of the most intriguing findings was the paradoxical relationship between elevated FC and slower eGFR decline, which contrasts with previous findings showing that calprotectin levels in urine and plasma correlate with adverse renal outcomes [11,41]. Felix et al. [11] reported that urinary calprotectin, along with NGAL and KIM-1, was a strong predictor of CKD progression. Similarly, Bourgonje et al. [41] found that plasma calprotectin was associated with an increased risk of new-onset CKD in the general population. These findings raise the possibility that the role of calprotectin may vary depending on its biological compartment (i.e., urine, plasma, or feces), warranting further investigation.

Several hypotheses may explain the unexpected association between elevated FC levels and renal outcomes. First, elevated FC levels may reflect an acute inflammatory response that, while typically harmful in the long term, may have transient protective effects. Acute inflammation contributes to the neutrophil-mediated clearance of pathogens and cellular debris, and prior studies suggest that neutrophils may play dual roles in both tissue injury and repair in kidney damage [42,43]. Second, elevated FC levels may indicate an adaptive immune response aimed at restoring intestinal barrier integrity. In CKD, intestinal dysfunction and increased gut permeability lead to the translocation of gut-derived toxins, exacerbating systemic inflammation [30,31]. Elevated FC levels may reflect an immune-mediated attempt to counteract these effects, temporarily reducing systemic inflammation and slowing renal decline. Third, the role of gut microbiota in CKD progression may be more complex than previously thought. Research has shown that uremic toxin production by gut bacteria does not follow a linear pattern, suggesting that intestinal dysbiosis and its effects on renal function vary across stages of CKD [44]. Finally, FC levels may fluctuate over time, and a single measurement may not accurately capture FC’s role in CKD progression. Given that inflammation and gut permeability change dynamically throughout the disease’s course, serial FC measurements may be necessary to fully understand its long-term impact on renal outcomes [37,39].

The association between elevated FC levels and increased mortality risk observed in our study aligns with prior research linking calprotectin levels to adverse outcomes in CKD populations. Kanki et al. [45] demonstrated that serum calprotectin independently predicted mortality in hemodialysis patients, particularly those with high phosphatemia, suggesting a potential link between mineral metabolism, inflammation, and cardiovascular risk in CKD. Our study expands this evidence to non-dialysis CKD populations, revealing that FC may serve as a prognostic marker for mortality beyond plasma and serum calprotectin. Subgroup analyses revealed a stronger association between FC and mortality in males, elderly patients (≥75 years), and non-diabetic CKD patients, suggesting demographic and comorbidity-specific differences in FC’s prognostic significance. In contrast, previous studies in diabetic populations, such as those by Winther et al. [46], Fedulovs et al. [38], and Lassenius et al. [47], have reported elevated FC levels in type 1 diabetes patients with macroalbuminuria or progressive diabetic kidney disease (DKD), suggesting a distinct gut–kidney axis in diabetes. These differences may reflect variations in gut microbiota composition, intestinal barrier integrity, and inflammatory responses, emphasizing the need for personalized interpretation of FC based on patient characteristics.

Our findings contribute to the growing body of evidence supporting the gut–kidney axis as a key modulator of systemic inflammation and CKD progression. FC, a cytosolic protein complex released by activated neutrophils and monocytes, is classically regarded as a marker of intestinal inflammation. Elevated FC levels were associated with higher hs-CRP, hypoalbuminemia, and anemia, aligning with malnutrition–inflammation–atherosclerosis (MIA) syndrome, as recently characterized in CKD patients by Allawi et al. [48]. These findings suggest that FC may reflect not only localized intestinal inflammation, but also systemic inflammatory burden in CKD.

The interpretation of FC in CKD requires careful consideration of multiple confounding factors that influence inflammation. The relationship between FC and systemic inflammation is complex and bidirectional. In our study, we addressed potential confounding by adjusting all analyses for multiple factors, including age, albumin, malignancy, total protein, coronary artery disease, blood urea nitrogen, creatinine, history of AKI, and hs-CRP. Despite these adjustments, residual confounding may persist due to the multifactorial nature of systemic and intestinal inflammation in CKD. Uremia-associated alterations represent primary confounders in the interpretation of FC. Gut microbiota dysbiosis and chronic systemic inflammation disrupt intestinal epithelial homeostasis, increase gut permeability, and facilitate neutrophil trafficking into the lumen, leading to elevated FC secretion [29,33]. This “leaky gut” phenomenon is largely mediated by pro-inflammatory cytokines, including interleukin-1β (IL-1β), IL-6, and TNF-α (tumor necrosis factor-α), which impair tight junction integrity and compromise the mucosal barrier [49,50,51,52]. Recent clinical studies have confirmed that CKD patients exhibit tight junction abnormalities and increased intestinal permeability, even in the absence of overt gastrointestinal disease [53]. Moreover, external factors such as the use of proton pump inhibitors [54], and comorbidities like diabetes and cardiovascular disease, may further aggravate gut barrier dysfunction and confound the interpretation of FC levels [33,53,55].

In addition, uremic toxins such as indoxyl sulfate and p-cresyl sulfate—protein-bound metabolites derived from the microbial metabolism of dietary amino acids—are well-established contributors to CKD progression. These toxins induce intestinal oxidative stress and immune activation, potentially amplifying gut barrier impairment [56,57,58]. Other gut-derived metabolites such as trimethylamine N-oxide (TMAO), phenylacetylglutamine, and hippuric acid have been implicated in both renal damage and cardiovascular complications, reflecting the systemic impact of gut dysbiosis [27,34]. These findings support the emerging concept that gut dysbiosis is a significant and independent contributor not only to renal outcomes, but also to cardiovascular complications and mortality in CKD patients [59,60]. Elevated levels of TMAO and indole derivatives have been associated with endothelial dysfunction, vascular calcification, and increased cardiovascular risk, which remains the leading cause of mortality in CKD [27,35,36]. Metabolomic analyses have further demonstrated strong correlations between the eGFR and circulating concentrations of these circulating metabolites, suggesting their potential utility as surrogate markers of renal function decline [28,29]. The interplay between gut dysbiosis, kidney dysfunction, and cardiovascular pathology underscores the systemic nature of CKD-related complications and supports an integrated organ axis model.

Microbiome-targeted therapies have also shown potential in mitigating CKD progression. Recent evidence suggests that supplementation with specific probiotic strains, such as *Lactobacillus johnsonii*, may help reverse CKD progression by modulating gut dysbiosis and reducing systemic inflammation [61]. Moreover, other *Lactobacillus* species have been shown to ameliorate experimental membranous nephropathy through immune modulation and gut barrier function restoration [62]. Furthermore, these interrelated mechanisms emphasize the complexity of using FC as a biomarker in CKD and highlight the need to interpret its levels within the broader framework of gut–kidney axis dysfunction. Future research should incorporate simultaneous measurements of serum and fecal calprotectin, microbiome analysis, and intestinal permeability markers to clarify the source of FC elevation and its clinical significance in CKD [29,33,53].

FC offers a more direct, non-invasive assessment of intestinal inflammation by reflecting neutrophil activity in the gut [33,49,50]. Although more costly and less routinely used, FC may provide complementary insights, particularly in CKD patients where gut barrier disruption and microbial translocation play a key role in systemic inflammation [29,52,53]. The relationship between inflammation and FC in CKD requires careful evaluation due to potential confounding factors. Although FC is primarily recognized as a non-invasive marker of intestinal inflammation, systemic inflammatory responses characteristic of CKD may also influence its levels. Traditional inflammatory markers such as CRP and serum albumin are widely used in clinical practice due to their accessibility and low cost. While they lack specificity for intestinal inflammation, they may still reflect gut-derived immune activation indirectly through systemic inflammation in CKD [51,63]. The routine clinical value of FC remains uncertain, but combining it with conventional markers like CRP may enhance risk stratification.

Our combined analysis of FC and CRP supports this complementary role and further strengthens the prognostic significance of inflammation in CKD. Patients with high FC and high CRP levels had the highest mortality risk, reinforcing the role of inflammation in CKD progression. Interestingly, patients with low FC but high CRP levels had relatively better survival outcomes than those with medium or high FC levels and low CRP. This suggests that FC captures distinct inflammatory signals not reflected by systemic markers alone, highlighting the heterogeneity of systemic inflammation in CKD and the need for multiple biomarkers to achieve accurate risk stratification.

Our findings have several important clinical implications and benefits. Our study highlights the potential of FC as a complementary inflammatory marker and contributes valuable data to the limited literature on FC in CKD, particularly its relationship with systemic inflammation and mortality. The paradoxical association between FC and eGFR outcomes warrants further investigation and may inform the timing of therapeutic interventions targeting intestinal inflammation in CKD. The combined assessment of FC and CRP may enhance risk stratification, identifying high-risk CKD patients who may benefit from intensive monitoring and management. Subgroup analyses further provide insights into demographic and comorbidity-related influences on FC’s prognostic significance, distinguishing our study from prior research primarily focused on urinary, plasma, or serum calprotectin.

However, several limitations must be acknowledged. The cross-sectional design limits causal inferences regarding the relationship between FC, systemic inflammation, and outcomes. The single-center nature of our study may reduce generalizability to broader CKD populations. Potential confounding factors, such as proton pump inhibitor use, which can elevate FC levels [54], were not fully accounted for. The absence of longitudinal FC measurements prevents the assessment of how changes in FC levels over time relate to disease progression. Finally, the lack of comprehensive gut microbiota and gut permeability analysis constrains the interpretation of specific mechanisms linking FC, gut dysbiosis, and systemic inflammation.

To address these limitations, future research should prioritize longitudinal designs to monitor FC trends, alongside mechanistic studies on neutrophil signaling, gut barrier function, and microbial composition. Comparative studies examining calprotectin levels across serum, urine, and feces are also needed to clarify compartment-specific roles. Finally, cost-effectiveness analyses will be essential to determine the feasibility of incorporating FC testing into routine CKD management.

## 5. Conclusions

In conclusion, our study provides evidence supporting fecal calprotectin as a clinically promising biomarker of systemic inflammation in CKD patients, with significant associations with mortality risk and potential implications for renal outcomes. The combined assessment of FC and CRP enhances prognostic evaluation, reinforcing the importance of the gut–kidney–inflammation axis in CKD. While further research is needed to fully elucidate underlying mechanisms and establish clinical utility, our findings suggest that FC monitoring may contribute to improved risk stratification and potentially guide therapeutic interventions targeting gut health in CKD patients.

## Figures and Tables

**Figure 1 biomolecules-15-00557-f001:**
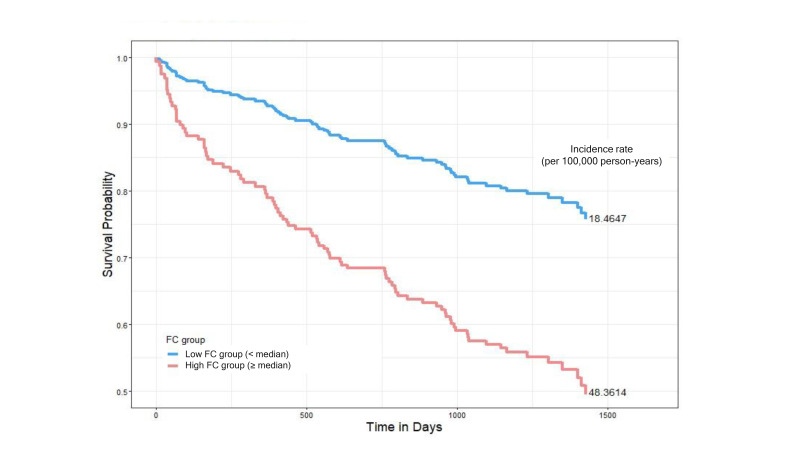
Kaplan–Meier survival curves comparing all-cause mortality between low- and high-fecal-calprotectin (FC) groups. Patients were stratified based on the median FC value. The low-FC group (<median, blue line) had a mortality incidence of 18.46 per 100,000 person-years, while the high-FC group (≥median, red line) showed a higher mortality incidence of 48.36 per 100,000 person-years. Hazards ratios were estimated using Cox proportional hazards models.

**Figure 2 biomolecules-15-00557-f002:**
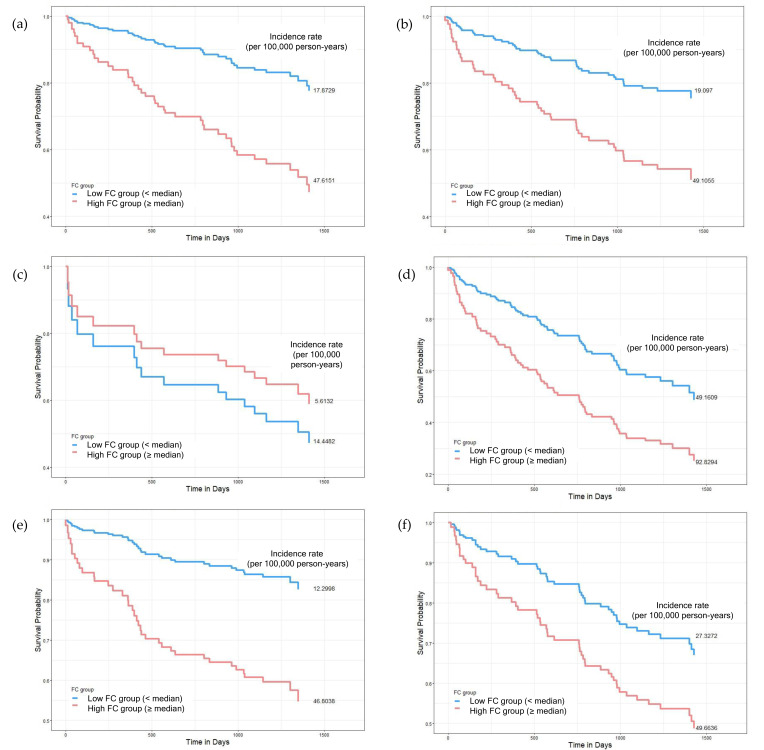
Subgroup Kaplan–Meier survival curves comparing mortality according to fecal calprotectin (FC) levels across sex, age, and diabetes status. Patients were divided into low- (<median, blue) and high-FC groups (≥median, red). Subgroups include (**a**,**b**) sex (male vs. female), (**c**,**d**) age (<75 vs. ≥75 years), and (**e**,**f**) diabetes status (non-diabetic vs. diabetic). In most subgroups, higher FC levels were associated with reduced survival, particularly in males (**a**), older adults (**d**), and non-diabetic patients (**e**). Interestingly, in individuals under 75 years (**c**), the low-FC group had higher mortality. Mortality incidence rates (per 100,000 person-years) are reported. Cox regression was used for hazards estimation.

**Figure 3 biomolecules-15-00557-f003:**
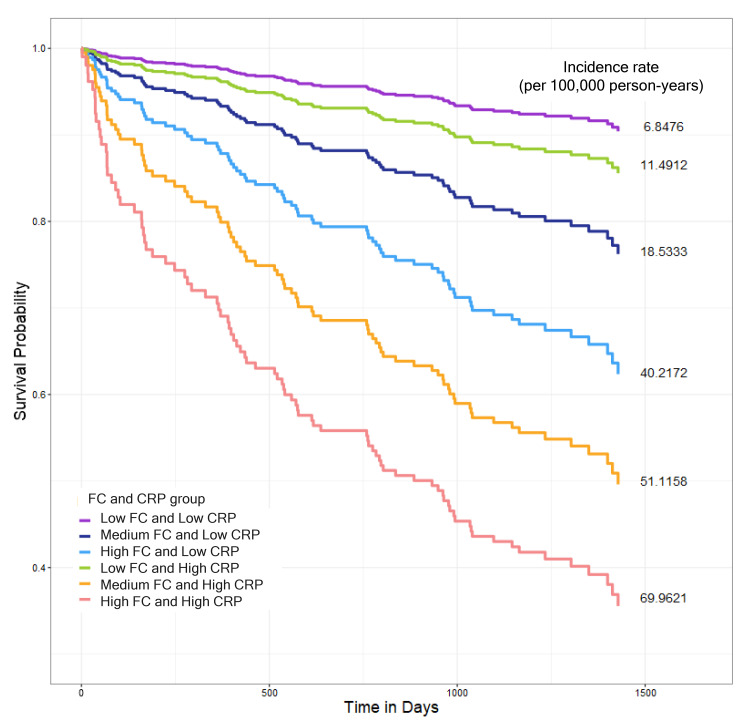
Kaplan–Meier survival curves showing all-cause mortality outcomes based on combined fecal calprotectin (FC) and high-sensitivity C-reactive protein (hs-CRP) levels. FC levels were categorized as low (<30 µg/g), medium (≥30 to <300 µg/g(), or high (≥300 µg/g), and hs-CRP levels were classified as low (<5 mg/L) or high (≥5 mg/L). Based on these cutoffs, patients were stratified into six groups: low FC and low CRP (purple), medium FC and low CRP (navy blue), high FC and low CRP (light blue), low FC and high CRP (green), medium FC and high CRP (orange), and high FC and high CRP (red). The high-FC and-high CRP group exhibited the highest all-cause mortality rate (69.96 per 100,000 person-years). The low-FC and high-CRP group (green line) demonstrated lower mortality rates (11.49 per 100,000 person-years) compared to those with the medium-FC and low-CRP group (18.53) or high-FC and low-CRP group (40.22). Survival differences among groups were visualized using Kaplan–Meier curves, and hazards ratios were estimated using Cox proportional hazards models adjusting for potential confounders.

**Figure 4 biomolecules-15-00557-f004:**
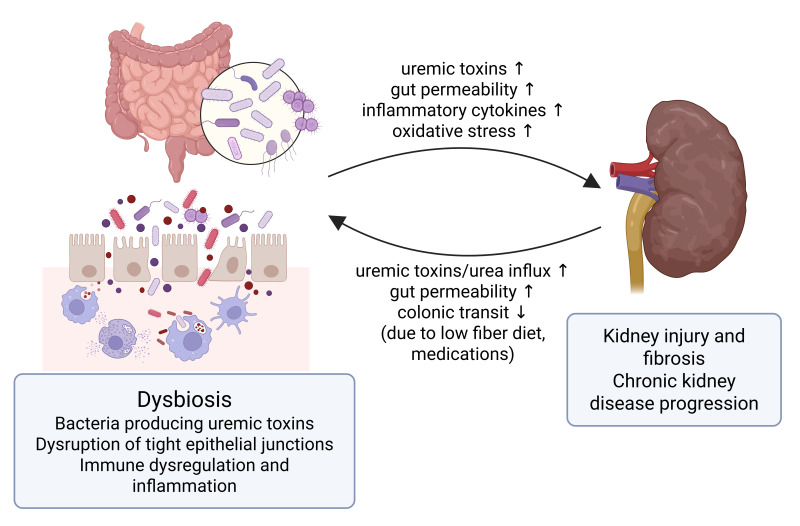
The gut–kidney axis in chronic kidney disease (CKD). This figure illustrates the bidirectional interplay between the gut and kidney in CKD. Dysbiosis leads to increased production of gut-derived uremic toxins, disruption of tight epithelial junctions, and heightened intestinal permeability. These changes promote microbial translocation, immune dysregulation, and systemic inflammation, which contribute to kidney injury, fibrosis, and CKD progression. Conversely, CKD-related factors such as delayed colonic transit, which is linked to low fiber intake and certain medications, and increased urea influx into the gut further exacerbate gut barrier dysfunction, perpetuating the cycle of inflammation and organ damage. ↑ indicates increase; ↓ indicates decrease. The figure was created usinghttps://biorender.com (confirmation of publication and licensing rights agreement number: SL284K06N6).

**Table 1 biomolecules-15-00557-t001:** Basal characteristics of study participants stratified by fecal calprotectin levels.

	Total (*n* = 260)	FC < Median (*n* = 130)	Median ≤ FC (*n* = 130)	*p*-Value
Fecal calprotectin, µg/g	136.22 ± 112.84	38.86 ± 12.55	233.58 ± 79.4	<0.0001 *
Age, year	68.41 ± 17.03	64.02 ± 18.15	72.8 ± 14.63	<0.0001 *
Female, *n* (%)	130 (50)	64 (49.23)	66 (50.77)	0.8041
Comorbidity				
DM, *n* (%)	127 (48.85)	55 (42.31)	72 (55.38)	0.0349 *
HTN, *n* (%)	184 (70.77)	93 (71.54)	91 (70)	0.7851
Malignancy, *n* (%)	37 (14.23)	16 (12.31)	21 (16.15)	0.3748
Glomerular disease, *n* (%)	42 (16.15)	30 (23.08)	12 (9.23)	0.0024 *
Heart failure, *n* (%)	38 (14.62)	10 (7.69)	28 (21.54)	0.0016 *
CAD, *n* (%)	20 (7.69)	6 (4.62)	14 (10.77)	0.0626
Chronic liver disease, *n* (%)	13 (5)	6 (4.62)	7 (5.38)	0.776
History of AKI, *n* (%)	68 (26.15)	24 (18.46)	44 (33.85)	0.0048 *
Medications				
Polystyrene sulfonate, *n* (%)	65 (25)	34 (26.15)	31 (23.85)	0.6674
AST-120, *n* (%)	6 (2.31)	3 (2.31)	3 (2.31)	1.0000
Ferrous sulfate, *n* (%)	47 (18.08)	20 (15.38)	27 (20.77)	0.2593
Laboratory data				
Positive stool OB, *n* (%)	27 (10.38)	2 (1.54)	25 (19.23)	<0.0001 *
BUN, mg/dL	42.1 ± 31.98	40.14 ± 31.59	44.06 ± 32.37	0.3236
Creatinine, mg/dL	2.87 ± 3.09	3.07 ± 3.23	2.67 ± 2.94	0.3058
eGFR, mL/min/1.73 m^2^	50.88 ± 44.43	50.48 ± 43.25	51.28 ± 45.74	0.8846
Hb, g/dL	10.77 ± 2.29	11.15 ± 2.32	10.39 ± 2.2	0.0073 *
Hematocrit, %	31.46 ± 6.55	32.61 ± 6.6	30.3 ± 6.32	0.0043 *
Total protein, g/dL	6.21 ± 0.93	6.41 ± 0.92	6 ± 0.91	0.0003 *
Albumin, g/dL	3.37 ± 0.72	3.55 ± 0.69	3.19 ± 0.7	<0.0001 *
Sodium, mmol/L	136.37 ± 6.87	136.47 ± 5.68	136.28 ± 7.9	0.8218
Potassium, mmol/L	4.2 ± 0.84	4.31 ± 0.89	4.09 ± 0.79	0.032 *
Hs-CRP, mg/L	6.77 (1.32, 46.57)	2.61 (0.67, 13.24)	18.6 (3.33, 86.61)	0.0032 *
Urine volume (UV), mL	1538.1 ± 902.53	1632.95 ± 982.19	1443.26 ± 807.83	0.0902
Total protein (urine), mg/dL	65 (23, 189.15)	75.75 (27.2, 208.7)	54.95 (22.4, 168.5)	0.9179
24 h Urine protein, mg/day	774.36 (251.73, 2257.33)	870.39 (243.91, 2534)	592.33 (254.14, 1751.2)	0.5134

Values are presented as mean ± standard deviation, number (percentage), or median (interquartile range). Abbreviations: FC = fecal calprotectin; DM = diabetes mellitus; HTN = hypertension; CAD = coronary artery disease; AKI = acute kidney injury; OB = occult blood; BUN = blood urea nitrogen; eGFR = estimated glomerular filtration rate; Hb = hemoglobin; Hs-CRP = high-sensitivity C-reactive protein. * *p* < 0.05.

**Table 2 biomolecules-15-00557-t002:** Multivariate logistic regression analysis for factors associated with ESKD development.

	OR	95% CI	*p*-Value
Fecal calprotectin group (≥median)	1.289	0.455–3.650	0.6323
Hemoglobin	6.034	1.719–21.183	0.0050 *
Hematocrit	0.611	0.401–0.930	0.0217 *
Urea nitrogen	1.034	1.009–1.060	0.0076 *
Creatinine	0.211	0.121–0.368	<0.0001 *
Total protein	2.393	1.312–4.366	0.0045 *
Hs-CRP	1.475	1.123–1.936	0.0052 *
Urine volume	1.001	1.000–1.001	0.0248 *

Analysis performed using stepwise model selection with Akaike Information Criterion (*n* = 119). Abbreviations: ESKD = end-stage kidney disease; OR = odds ratio; CI = confidence interval; Hs-CRP = high-sensitivity C-reactive protein. * *p* < 0.05.

**Table 3 biomolecules-15-00557-t003:** Multivariate logistic regression analysis for factors associated with eGFR change rate.

	OR	95% CI	*p*-Value
Fecal calprotectin group (≥median)	2.763	1.139–6.699	0.0245 *
History of AKI	0.295	0.131–0.666	0.0033 *
Hs-CRP	0.605	0.483–0.757	<0.0001 *

Analysis performed using stepwise model selection with Akaike Information Criterion (*n* = 174). Abbreviations: OR = odds ratio; CI = confidence interval; eGFR = estimated glomerular filtration rate; AKI = acute kidney injury; Hs-CRP = high-sensitivity C-reactive protein. * *p* < 0.05.

**Table 4 biomolecules-15-00557-t004:** Cox proportional hazards regression analysis for predictors of all-cause mortality.

	HR	95% CI	*p*-Value
Fecal calprotectin group (≥median)	1.658	1.034–2.658	0.0357 *
Age	1.061	1.034–1.088	<0.0001 *
Albumin	0.304	0.182–0.507	<0.0001 *
Malignancy	1.872	1.113–3.147	0.0179 *
Total protein	1.592	1.148–2.208	0.0052 *
CAD	3.506	1.739–7.071	0.0004 *
BUN	1.018	1.007–1.029	0.0012 *
Creatinine	0.8	0.669–0.956	0.0141 *
History of AKI	0.55	0.327–0.924	0.0241 *
Hs-CRP	1.131	0.979–1.305	0.0926

Analysis stratified by enrollment date (*n* = 260). Abbreviations: HR = hazards ratio; CI = confidence interval; CAD = coronary artery disease; BUN = blood urea nitrogen; AKI = acute kidney injury; Hs-CRP = high-sensitivity C-reactive protein. * *p* < 0.05.

**Table 5 biomolecules-15-00557-t005:** Subgroup analyses for the association between fecal calprotectin and mortality.

Subgroup	HR	95% CI	*p*-Value	Incidence Rates **
Sex				
Male (*n* = 130)	2.16	1.046–4.463	0.0375 *	47.62 vs. 17.87
Female (*n* = 130)	1.301	0.667–2.538	0.4403	49.11 vs. 19.10
Age				
<75 years (*n* = 139)	0.828	0.224–3.066	0.7772	5.61 vs. 14.45
≥75 years (*n* = 121)	2.122	1.209–3.725	0.0088 *	92.83 vs. 49.16
Diabetes status				
Non-DM (*n* = 133)	2.487	1.141–5.421	0.0219 *	46.80 vs. 12.30
DM (*n* = 127)	1.194	0.603–2.362	0.611	49.66 vs. 27.33

All analyses adjusted for age, albumin, malignancy, total protein, coronary artery disease, blood urea nitrogen, creatinine, history of AKI, and hs-CRP. Abbreviations: HR = hazards ratio; CI = confidence interval; DM = diabetes mellitus; AKI = acute kidney injury; hs-CRP = high-sensitivity C-reactive protein. * *p* < 0.05. ** Incidence rates per 100,000 person-years for high- (≥median) vs. low-fecal-calprotectin groups (<median).

**Table 6 biomolecules-15-00557-t006:** Cox regression analysis for the combined effect of fecal calprotectin and CRP on mortality.

Variable	HR	95% CI	*p*-Value
FC and CRP			<0.0001 *
Low FC and low CRP	1.000 (reference)	-	-
Medium FC and low CRP	1.528	0.491–4.750	
High FC and low CRP	2.147	0.588–7.838	
Low FC and high CRP	0.383	0.079–1.860	
Medium FC and high CRP	2.584	0.876–7.626	
High FC and high CRP	3.504	1.163–10.554	
Other variables			
Age	1.076	1.050–1.103	<0.0001 *
Albumin	0.382	0.234–0.623	0.0001 *
Polystyrene sulfonate	1.670	1.044–2.670	0.0323 *
Urine volume	1.000	0.999–1.000	0.0123 *
24 h urine protein	1.280	0.920–1.781	0.1432
History of AKI	0.792	0.492–1.274	0.3360

Analysis stratified by enrollment date (*n* = 260). FC categories: low (FC < 30 μg/g); medium (30 µg/g ≤ FC < 300 µg/g); high (FC ≥ 300 μg/g). CRP categories: low (CRP < 5 mg/L); high (CRP ≥ 5 mg/L). Abbreviations: FC = fecal calprotectin; CRP = C-reactive protein; HR = hazards ratio; CI = confidence interval; AKI = acute kidney injury. * *p* < 0.05.

## Data Availability

The data presented in this study are available in the article. Further inquiries can be directed to the corresponding author.

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
