# Peer review of "Fecal Calprotectin as a Prognostic Biomarker for Mortality and Renal Outcomes in Chronic Kidney Disease"

_biomolecules, 2025, doi:10.3390/biom15040557_

Round 1
Reviewer 1 Report
Comments and Suggestions for Authors
In this work, the authors reveal fecal calprotectin as a prognostic biomarker for mortality and renal outcomes in chronic kidney disease. Several suggestions are made as follows to improve the quality of the manuscript.
- The abstract should be improved. Please more result in details in abstract.
- Abbreviations should be avoided in keywords. Please provide more keywords for the manuscript. In addition, cardiovascular disease should be presented as a keyword.
- The prevalence or Incidence or mortality or morbidity of CKD should be introduced in the paragraph 1 of introduction section.
- Please remove reference 1-4 or cite the latest references in the introduction section.
- The authors described that “CKD patients frequently exhibit heightened intestinal inflammation and significant alterations in gut microbiota composition [42-45]”. The dysbiosis of gut microbiota in CKD was reported by several publications. In addition, the latest several studies reported the targeting Lactobacillus johnsonii reverse chronic kidney disease and Lactobacillus species ameliorate membranous nephropathy. Please discuss more studies to improve manuscript.
- The authors described that “the overgrowth of pathogenic bacteria leads to increased production of uremic toxins, particularly indoxyl sulfate and p-cresyl sulfate [46-48]. These toxins, combined with compromised intestinal barrier integrity, promote systemic inflammation and accelerate CKD progression [49-55]”. Increasing publications have shown that a variety of uremic toxins were identified in CKD. recently, a number of metabolites including uremic toxins identified from CKD patients were demonstrated to mediate renal fibrosis reported by recent and latest publications. These studies uncovered that the associations between renal function and circulating metabolites including uremic toxins in CKD patients. Please discuss more studies to improve manuscript.
- The authors described that “Recent studies have demonstrated that altered gut microbiota is independently associated with cardiovascular disease and increased mortality in CKD patients [56,57]”. The reviewer suggests that the authors discuss more to improve this hot topic in the manuscript.
- Please improve the discussion.
- Please provide more details for figure legends.
- There are some very old references such as 1-4, 7, 18, 20, 22, 27, 29, 35-37, 40, 42, 62, 63, 66, 68, 73. Please cite and update the latest publications. Please cite the recent 10 years references.
- Please provide high-resolution images for figures.
- The language should be improved or polished by English native speaker.
The language should be improved or polished by English native speaker.
Author Response
Thank you for the opportunity to revise our manuscript. We are grateful to the reviewer for their constructive comments, which we believe have helped improve the quality and clarity of our paper. We have made every effort to address all the comments and incorporate the reviewer’s suggestions. Corresponding revisions have been highlighted in the revised manuscript using track changes. Please find the detailed point-by-point responses in the attached response file.

Reviewer 2 Report
Comments and Suggestions for Authors
Dear Authors, thanks for this manuscript which is really interesting. The object of the paper is quite original and its novelty is confirmed in literature. The gut-kidney axis could be explained with some figures. Materials and methods are appropriated. The role of inflammation and its influence on FC should be enlarged and deeply investigated to avoid confoundind factors. Moreover the opportunity to routinarily evaluate FC should be explained because inflammation markers are already available and cheaper.
Author Response

(The authors gave the same response as above.)

Round 2
Reviewer 1 Report
Comments and Suggestions for Authors
I suggest that the manuscript is published.